# Down-Regulation of Photosynthetic Electron Transport and Decline in CO_2_ Assimilation under Low Frequencies of Pulsed Lights

**DOI:** 10.3390/plants10102033

**Published:** 2021-09-28

**Authors:** Marguerite Cinq-Mars, Guy Samson

**Affiliations:** Département des Sciences de l’Environnement, Groupe de Recherche en Biologie Végétale, Université du Québec à Trois-Rivières, Trois-Rivières, QC G8Z 4M3, Canada; Marguerite.Cinq-Mars@uqtr.ca

**Keywords:** photosystem, cyclic electron flow, photoprotection, nonphotochemical quenching, LEDs

## Abstract

The decline in CO_2_ assimilation in leaves exposed to decreasing frequencies of pulsed light is well characterized, in contrast to the regulation of photosynthetic electron transport under these conditions. Thus, we exposed sunflower leaves to pulsed lights of different frequencies but with the same duty ratio (25%) and averaged light intensity (575 μmoles photons m^−2^ s^−1^). The rates of net photosynthesis P*n* were constant from 125 to 10 Hz, and declined by 70% from 10 to 0.1 Hz. This decline coincided with (1) a marked increase in nonphotochemical quenching (NPQ), and (2) the completion after 25 ms of illumination of the first phase of P_700_ photooxidation, the primary electron donor of PSI. Under longer light pulses (<5 Hz), there was a slower and larger P_700_ photooxidation phase that could be attributed to the larger NPQ and to a resistance of electron flow on the PSI donor side indicated by 44% slower kinetics of a P_700_^+^ dark reduction. In addition, at low frequencies, the decrease in quantum yield of photochemistry was 2.3-times larger for PSII than for PSI. Globally, our results indicate that the decline in CO_2_ assimilation at 10 Hz and lower frequencies coincide with the formation of NPQ and a restriction of electron flows toward PSI, favoring the accumulation of harmless P_700_^+^.

## 1. Introduction

Steady-state photosynthesis is the exception rather than the norm for plants of forest and crop understories. In these environments, most of the available light arrives as sunflecks, i.e., periods of high light intensity typically shorter than 10 s, and occurs at frequencies ranging from 0.1 Hz to 30 Hz [1,2]. Photosynthesis responds to these light fluctuations by various regulatory mechanisms acting on different timescales to control light-harvesting (e.g., nonphotochemical quenching NPQ), electron flows (e.g., cyclic electron flow around Photosystem I, CEF-PSI), and enzymatic activities (e.g., thioredoxin-mediated enzyme activation) [3]. However, the delays between these photosynthetic responses and light fluctuations may be responsible for a 20–30% loss in plant productivity [4,5,6,7].

To better understand the dynamics of photosynthesis under fluctuating light, a valuable alternative to the random sunflecks of natural light is the use of repetitive variations in light intensities, such as pulsed lights [8,9,10]. These lights were used in seminal studies such as Emerson and Arnold [8] who were the first to correctly describe the dynamic interaction between the photochemical and biochemical reactions of photosynthesis. By measuring CO_2_ assimilation in *Chlorella* sp. under intermittent lights, they deduced that at the end of the light period, the photochemical reactions cease immediately but the biochemical reactions continue until the photochemical products are consumed by the biochemical reactions, with no other reactions possible until further illumination. This principle underlies the recent kinetic model introduced by Jishi et al. [9,11] explaining the relation between net photosynthetic rates P*n* of cos lettuce leaves and the characteristics of pulsed light: P*n* declines at low frequencies and/or low duty ratios (i.e., the proportion of pulse duration over a cycle), and this decline is more pronounced at higher averaged PPFDs (photosynthetic photon flux densities). Indeed, longer and more intense light pulses completely reduce the pool of photosynthetic intermediates, and longer dark intervals cause their depletion. In their model, the functioning of photosynthesis under pulsed lights could be satisfactorily explained by the dynamics of production and consumption of photosynthetic intermediates, even though the photochemical efficiency of photosystem II (PSII) was assumed to be unaffected by pulsed lights of low frequencies and duty ratios that limit photosynthesis.

However, it is now well established that the functioning of photosynthetic electron transport under fluctuating lights is different than under steady-state conditions [12,13,14]. After a prolonged period of low light intensity where the photoprotective mechanisms are less activated, a sudden increase in light intensity causes a burst of electrons to PSI, which rapidly reduce its electron acceptors, thereby enhancing the O_2_ photoreduction and formation of reactive oxygen species [3,15]. Under repetitive short pulses of light, the cumulative effects of these reactions are responsible for the higher susceptibility of PSI to photoinhibition relative to PSII, in contrast to the well-known PSII photoinhibition observed under high-constant light [16]. Results from Miyake’s lab demonstrated that repetitive light pulses (300 ms and 20,000 μmoles m^−2^ s^−1^) for 4 h significantly decreased the rates of CO_2_ assimilation and the amount of photooxidable P_700_, the primary electron donor of photosystem I [16,17,18]. They showed that during these intense light pulses, P_700_ remained mostly reduced, whereas in the presence of a strong background constant light (1400 μmoles m^−2^ s^−1^), P_700_ was largely oxidized during the light pulses and the PSI photoinhibition was avoided. These results demonstrated the importance of maintaining P_700_ in an oxidized state to avoid PSI photoinhibition, not only under pulsed lights but in any situations where the Calvin–Benson cycle is suppressed.

Several regulatory mechanisms contribute to maintain P_700_ oxidized under fluctuating light [18]. Among them, the rapid activation of the cyclic electron flow around PSI (CEF-PSI) mediated by the stromal NADPH dehydrogenase or PGR5 proteins is particularly effective to protect PSI against photoinhibition [15,17,19,20]. In contrast to the linear electron flow (LEF), the electrons during CEF-PSI are rapidly removed from the PSI acceptor side by returning to the intersystem electron transport chain, from reduced Fd to the plastoquinone (PQ) bound on the stromal side of Cyt b_6_f [21,22,23]. Then, the oxidation of PQH_2_ on the luminal side of the Cyt b_6_f contributes to the acidification of the lumen, required for ATP synthesis and the induction of the nonphotochemical quenching NPQ. For a given light intensity, this NPQ decreases the rate of PSII electron transport.

In addition, low luminal pH can, under some conditions, restrict electron flow in Cyt b6f, one of the mechanisms that contribute to the “photosynthetic control,” i.e., the coordination between the production of ATP and NADPH by the photochemical reactions with their utilization by the carbon metabolism [24]. The down-regulation of electron transport due to lumen acidification is easily observed in vitro [25,26,27], but its occurrence under physiological conditions is not universal. Resistance to linear electron flow (LEF) is often constant or even decreases with increasing light intensities under normal environmental conditions [28,29,30]. A higher resistance to LEF in Cyt b6f was shown under both high light and high or low CO_2_ levels, drought (and sorbitol-induced osmotic stress), and sucrose feeding [31,32]. Besides low luminal pH, there is evidence that down-regulation of the LEF in Cyt b_6_f may also depend on a redox-based mechanism, likely induced by a high degree of reduction of the NADP(H) pool, as indicated by the use of two transgenic tobacco (*Nicotiana tabacum*) lines, both with similar limitations of carbon reduction but differing by the redox state of the NADP(H) pool [33].

The presence of a down-regulation of the LEF in Cyt b_6_f is most frequently measured by the slower kinetics of P_700_^+^ reduction following a light-to-dark transition. Recently, Johnson and Berry [34] introduced a method for diagnosing the control of linear electron flow (LEF) based on chlorophyll-a fluorescence measurements. This method relies on the idea that the oxidation of PQH_2_ by Cyt b_6_f limits the steady-state rate of LEF, and that this oxidation has a first-order dependence on PQH_2_ [35]. Thus, the conductance (the inverse of resistance) to LEF at Cyt b_6_f can be obtained by the ratio of the rate of LEF over the redox state of the PQ pool estimated by the “1-qL” parameter [36]. Johnson and Berry observed that the rate of LEF increased linearly with the proportion of the PQH_2_/PQ pool (1-qL) as the light intensity increased. At and beyond saturating PFD, the LEF reached a maximum while the proportion of PQH_2_/(PQ + PQH_2_) continued to increase. Thus, the decrease in the LEF/(1-qL) ratio indicated a lower conductance of Cyt b_6_f to LEF under saturating light.

Some of the above-mentioned studies [8,9,10] indicate on the one hand that the decline in CO_2_ assimilation under pulsed lights of low frequencies and low duty ratios is caused by the complete reduction in a pool of electron acceptors during the light pulses. On the other hand, effective photoprotection against PSI photoinhibition under pulsed lights would involve a decrease in electron transport on the PSI donor side, increasing the proportion of P_700_^+^ [3,16,37]. Considering the uncertainties about the functioning of photosynthetic electron transport under pulsed lights, we examined in this study the regulation of electron transport at both PSI and PSII in relation to the rates of net photosynthesis P*n* measured under pulsed lights of moderate intensity (averaged PPFD of 575 μmoles m^−2^ s^−1^) given at frequencies ranging from 125 to 0.1 Hz. We measured a 70% decrease in the P*n* occurring from 10 to 0.1 Hz that coincided with (1) the formation of a significant nonphotochemical quenching (NPQ) with a corresponding decrease in PSII quantum yield, and (2) the increased resistance to electron transport indicated by a lower LEF/(1-qL) ratio and the marked slowdown of P_700_^+^ dark reduction. Both NPQ and the increased resistance to LEF at low frequencies contributed to enhance the accumulation of P_700_^+^ during the light pulses.

## 2. Results

### 2.1. Decline in Photosynthesis at Low Frequencies of Pulsed Lights

The rates of net photosynthesis P*n* were measured in sunflower leaves exposed to pulsed lights of different frequencies. Prior to the measurements, the leaves were illuminated for 20 min at a constant light intensity of 1000 μmol m^−2^ s^−1^ to allow the full induction of photosynthesis. Then, the leaves were exposed to pulsed lights consisting of 2300 μmol m^−2^ s^−1^ pulses at frequencies ranging from 125 Hz to 0.1 Hz, given in a random order, each one for about 8–10 min, allowing the reaching of a new “dynamic state.” The duty ratio of the pulsed lights was 25% (25% light on, 75% light off), so the averaged light intensity was 575 μmol m^−2^ s^−1^. Between each frequency, leaves were kept in darkness for 2 min to record the rates of dark respiration Rd (and to measure the minimal level of chlorophyll fluorescence Fo’, see below).

We observed that P*n* was constant from 125 Hz to 10 Hz (6.0 μmol CO_2_ m^−2^ s^−1^), i.e., when light pulses were equal to or shorter than 25 ms. From 10 Hz to 0.1 Hz, P*n* decreased by 70%, down to 1.8 μmol CO_2_ m^−2^ s^−1^ (Figure 1). In contrast, the rates of dark respiration R*d* measured between the periods of different light pulses were unaffected by the frequency. Thus, the rates of apparent photosynthesis (P*n* + R*d*) decreased by 53% from 125 Hz to 0.1 Hz.

### 2.2. Chlorophyll-a Fluorescence Transients and PSII Quantum Yields under Pulsed Lights

To better understand the relation between the decline in P*n* at low frequency and photosynthetic electron transport, we first examined the chlorophyll-a fluorescence (ChlF) transients during the light pulses given at different frequencies. From 125 Hz to 5 Hz, the ChlF signals were more or less constant during the light cycles, at least for the time resolution of our ChlF measuring system. At 2 Hz and lower frequencies, amplitudes of the ChlF variations during the light pulses increased, with ChlF rising from the lower levels, Fdark (black dots), between the pulses up to a peak ChlF level, Fp’, reached during the repetitive light pulses (blue dots) (Figure 2A). Even at the low frequencies, the long light pulses were not saturating. Indeed, from the low to high frequencies, the Fp’ levels represented only 41% (at high Hz) to 71% (at low Hz) of the maximum ChlF level, Fm’ (red dots), induced by a saturating pulse given just before the end of the pulsed light period (Figure 2A).

The decrease in Fm’ at frequencies lower than 10 Hz indicates the formation of a nonphotochemical quenching (NPQ) that should affect light utilization in PSII centers. Thus, we estimated the quantum yield of PSII photochemistry YII and the complementary quantum yields of regulated YNPQ and nonregulated YNO energy dissipation in PSII. These estimations rely on the measurement of ChlF yield F “*measured briefly before application of a Saturation Pulse*” [38]. The representative F level in pulsed light may be ambiguous due to the different amplitudes of ChlF variations during the repetitive light pulses, depending on the frequency of the pulsed lights. To minimize a possible bias in the estimation of F, we used the mean value of ChlF during a light pulse (F_pulse_) to calculate YII, YNPQ, and YNO (see Material and Methods). The results show that YII was constant from 125 Hz to 25 Hz and decreased by about 50% from 10 Hz to 0.1 Hz, a decrease that is compensated mainly by an increase in YNPQ, and to a lower extent by an increase in YNO (Figure 2B). The validity of the mean ChlF value during the light pulses for the calculation of the PSII quantum yields is supported by the correlation (R^2^ = 0.78) observed between YII and the rates of apparent photosynthesis (P*n* + R*d*), with the y-intercept close to zero (0.29) (Figure 3).

### 2.3. P_700_ Redox Kinetics and PSI Quantum Yields

The results presented above show that the decline in P*n* at low frequencies is associated with the formation of a large NPQ and not with an over-reduction of the PSII electron acceptors (lower Fp’ relative to Fm’), a condition that may limit electron donation to PSI. To verify this possibility, we analyzed the rapid transients of the P_700_ redox changes during the light pulses of different frequencies (Figure 4). The induction kinetics represent the changes in the proportion of photooxidized P_700_ (from 0 to 1) during one light cycle (25% light, 75% darkness). They are shown on different timescales to facilitate their comparison. At 0.1 Hz, the light pulses were sufficiently long to allow the induction of the first phases A to C of the typical P_700_ photooxidation kinetics following a dark–light transition, as identified by Harbinson and Headly [22]. At the beginning of the light pulses, there was a rapid phase (A) of P_700_ photooxidation that was completed within ≈25 ms. This corresponds to frequencies equal to or lower than 10 Hz, i.e., where P*n* became limited. The amplitude of this fast P_700_ photooxidation accounted for ≈13% of the total photooxidable P_700_. After this rapid photooxidation, there was a transient reduction phase (B) between 50 ms and 200 ms after the beginning of the pulses. This reduction appeared as a small inflection at 2 Hz, and became more pronounced (more P_700_ re-reduced) as the frequency increased. This second phase was then followed by a large and slow photooxidation phase (C), where about 27% P_700_ was photooxidized after about 1.6 s. This C level was reached only at a frequency of 0.1 Hz. As expected, P_700_ remained completely reduced between the light pulses. In addition, it is important to note that no PSI photoinhibition under pulsed lights was observed under our experimental conditions, as the extent of total P_700_ photooxidable (Pm − Po levels) remained constant during the measurements (see Appendix A).

Then, we estimated by the Saturation Pulse Method the quantum yield of PSI photochemistry (YI) and the quantum yields of energy dissipation due to limitations on its donor (YND) and acceptor (YNA) sides (Figure 5). This method was designed to assess the complementary PSI quantum yields under “*a given state of illumination*” [39]. As the P_700_ signal (P level) was continuously changing during the light pulses, we used its averaged values during the light pulses (as for ChlF) for the calculations of the quantum yields, and measured with a saturating Xe-pulse the maximum extent of P_700_ photooxidation under pulsed light (Pm’) and under far-red light (Pm). Compared to PSII, the complementary PSI quantum yields were less affected by the frequencies. YI decreased from 0.71 at 125 Hz to 0.61 at 0.1 Hz. The slopes estimated from logarithmic regressions indicated that this slight YI decrease can be explained by limitations mainly on the PSI donor side (75%), and secondarily on its acceptor side (17%), although the changes in YNA with the frequency were statistically insignificant.

### 2.4. Increased Resistance in the Intersystem Electron Chain under Low Frequencies of Pulsed Lights

The larger accumulation of P_700_^+^ observed under low frequencies of pulsed lights may result in part from an increased resistance in the intersystem electron chain. Thus, we used two independent approaches to assess this possibility. The first one is the new diagnostic method of Johnson and Berry [34] based on the estimations by ChlF of the rate of linear electron flow LEF (YII * “flux of absorbed light”) and the proportion of reduced plastoquinone PQH_2_ (approximated by 1-qL). As the steady-state rate of LEF is directly dependent on the concentration of PQH_2_, both LEF and “1-qL” should vary simultaneously unless there is a change in the conductance of Cyt b_6_f to LEF. The results presented in Figure 6 show that the LEF/(1-qL) was relatively constant at high frequencies (≥25 Hz) but markedly decreased at lower frequencies (≤10 Hz). A consequence of the lower conductance (or higher resistance) to LEF at Cyt b_6_f is a higher proportion of reduced PQH_2_ at low frequencies.

The second and more classical approach to detect an increased resistance to LEF at low frequencies of pulsed lights is the kinetic analysis of P_700_^+^ dark-reduction at the end of the light pulses [40]. For all frequencies, the P_700_^+^ reduction kinetics could be satisfactorily fitted (R^2^ > 0.98) to an exponential decay equation (y_0_ + a * e^−k*t^) from which the apparent first-order rate constant k was obtained (Figure 7). The kinetics of P_700_^+^ reduction monitored at the end of the light pulses were faster in leaves exposed to high frequencies than to low frequencies. More specifically, the rates k were constant at frequencies of 2 Hz and higher (246 s^−1^) and decreased by 42%, down to 143 s^−1^ at 0.2 Hz and 0.1 Hz (Figure 8).

## 3. Discussion

Previous studies have shown that pulsed lights can be as effective as constant light to support photosynthesis, but for a given duty ratio and averaged light intensity, the light use efficiency decreases at low frequencies [9,10]. In our experiments, the net photosynthetic rates P*n* in sunflower leaves were constant from 125 to 10 Hz and declined by 70% when the frequency of the pulsed lights decreased from 10 Hz to 0.1 Hz. The extent of this decline and the frequencies at which it occurred are similar to the observations made by Jishi et al. [9] with cos lettuce, who characterized how the frequency, duty ratio, and averaged light intensity of pulsed lights determine the decrease in the CO_2_ assimilation. The classical explanation for the loss of photosynthetic efficiency at low frequencies is the over-reduction of a pool of electron acceptors during the light pulses [8,9,10]. Indeed, at low frequencies, longer light pulses alternating with longer dark periods could cause an overreduction of the photosynthetic electron transport chain followed by a depletion of the photoproducts, thereby decreasing photosynthetic efficiency. In our experiments, the peak levels Fp’ of chlorophyll fluorescence measured during the repetitive light pulses were clearly lower than the maximum Fm’ level induced by a saturating pulse. This indicates that the light pulses were insufficiently intense and long to completely reduce the electron transport chain. We also observed that the decline in quantum yield of PSI photochemistry YI at low frequencies resulted mostly from a limitation on its donor side (lack of electrons). These observations indicated that an overreduction of a pool of electron acceptors during our light pulses is unlikely and cannot explain the decline in P*n* at low frequencies. However, we cannot rule out that the reduction in a critical fraction of a pool of electron acceptors (PQ, Fd, or NADP(H)) could be sufficient to cause the decline in photosynthetic efficiency.

Our results do not allow the clear identification of the primary factor responsible for the decline in photosynthesis under low frequencies of pulsed light. What the results clearly showed, however, is that this decline is associated with the induction of a photoprotective mode, characterized by the presence of NPQ and an increased resistance in the intersystem electron transport chain, both favoring the accumulation of P_700_^+^.

To better understand the down-regulation of the photosynthetic electron transport under fluctuating lights, the kinetic analyses of P_700_ redox changes are particularly informative. Under repetitive light pulses of duration ranging from 2 ms (125 Hz) to 2.5 s (0.1 Hz), we observed the development of the first three (A to C) phases of the typical P_700_ redox kinetics (phases A to F) following a dark–light transition [40,41]. From 125 to 10 Hz, during the light pulses, there was a progressive accumulation of P_700_^+^ reaching a plateau “A” after 25–30 ms of illumination. Interestingly, Schreiber [42] also observed this first plateau A of P_700_ photooxidation near 25 ms after the onset of a single 1400 μmol photons m^−2^ s^−1^ light pulse, and demonstrated that it coincides with the maximum oxidation of the electron donor to P_700_ plastocyanin (PC), and with an intermediary plateau of the biphasic reduction in the PSI electron acceptor ferredoxin (Fd). These transient redox changes in PC, P_700_, and Fd are considered as a clear manifestation of a fast cyclic electron flow around PSI (CEF-PSI), where, during this time window (between ≈25 ms and 50), the accumulation of Fd^−^ is postponed as it is rapidly oxidized by PQ molecules bound at the stromal side of Cyt b_6_f, which transfers electrons back to PC and P_700_^+^ (thereby limiting their oxidation). Similar observations were also simulated with the kinetic model of Belyaeva et al. [43]. Considering that (1) the phase A of the P_700_ photooxidation kinetics is associated with the induction of CEF-PSI [42,43]; (2) we observed this phase A completed after 25 to 50 ms of sudden illumination corresponding to light pulses of 10 to 5 Hz; (3) the photochemical efficiency of PSII YII decreased to a larger extent than YI as the frequencies decreased; and (4) CEF-PSI is stimulated under limiting conditions for CO_2_ assimilation and under the sudden increase in light intensities [31,44,45], it is therefore reasonable to suggest that the decline in P*n* that we observed at frequencies lower than 10 Hz is associated with the activation of a CEF-PSI at the expense of LEF supporting carbon metabolism.

Besides the removal of electrons on the PSI acceptor side, the main physiological function of CEF-PSI is to transfer protons into the lumen for ATP production and NPQ formation [21]. In addition, lumen acidification can activate the “photosynthetic control,” i.e., down-regulation of photosynthetic electron transport by the restriction of plastoquinol oxidation by the Cyt b_6_f [45]. This photosynthetic control can be detected in the P_700_ redox kinetics during the light pulses and immediately after (light-to-dark transition). Following the completion of the first phase (A) of P_700_^+^ of photooxidation, P_700_^+^ is re-reduced, as indicated by the dip B reached near 200 ms after the beginning of the pulses, when the electrons arriving from PSII and from the CEF-PSI reduce the intersystem electron carriers and P_700_^+^. We observed that the magnitude of the dip B increased as the frequency of pulsed light decreased from 2 Hz to 0.1 Hz. In the case of a single dark-to-light transition, the depth of the B transient increases with the duration of the darkness before the illumination [41] as it is caused by (1) a rush of electrons from PSII on the PSI donor side rapidly reducing P_700_^+^, associated with (2) a limitation of electron transfer on the PSI acceptor side due to the slow light-activation of the ferredoxin-NADP^+^ reductase and of the carbon reduction cycle enzymes. Thus, in the case of a single dark-to-light transition, the large and slow P_700_ photooxidation corresponding to the transients from B to C (up to F) is related to the restoration of the electron transfer on the PSI acceptor side due to light-activation of stromal enzymes [24,40].

However, these interpretations for the P_700_ redox kinetics during a single dark–light transition probably differ from those for P_700_ kinetics observed under repetitive light pulses. Indeed, the stromal enzymes should be sufficiently activated under our light conditions (20 min pre-illumination of 1000 μmol m^−2^ s^−1^), and at a relatively constant level during a dark–light cycle. Thus, the deepening of the dip B observed as the frequencies decreased from 2 Hz to 0.1 Hz and the following large accumulation of P_700_^+^ (level C) during the relatively long light pulses (1.25 and 2.5 s) rather indicate a slower rate of electron transfer coming from PSII at low frequencies due to both (1) a lower quantum yield of PSII photochemistry YII associated with a larger nonphotochemical quenching YNPQ, and (2) a higher resistance in the intersystem electron transport chain between PSII and PSI. Besides the increase in YND at lower frequencies (Figure 5), a reliable manifestation of this down-regulation of electron transport is the slower P_700_^+^ reduction at the end of the light pulses (light-to-dark transition) (see [40] and references therein). From 125 Hz to 2 Hz, the P_700_^+^ dark-reduction rates were constant (238 s^−1^) and fell by 40% (142 s^−1^) from 2 to 0.1 Hz. These observations are supported by the decrease in Cyt b_6_f conductance to LEF estimated by the parameter LEF/(1-qL) [34]. The decrease in LEF/(1-qL) was already noticeable at 10 Hz and was larger (76%) than the decrease in P_700_^+^ reduction rates. The higher sensitivity to low frequencies of LEF/(1-qL) compared to the P_700_^+^ dark-reduction rates may be explained by methodological differences as the changes in LEF/(1-qL) and YNPQ, both estimated by chlorophyll fluorescence, are closely mirrored (Figure 2A and Figure 6). It has also been shown that in presence of zeaxanthin (as in illuminated normal leaf), the formation of NPQ could occur at significantly higher luminal pH (higher pK) than the inhibition of PQH_2_ oxidation [32]. Nishio and Withmarsh [26] observed that the Cyt f reduction rate remained maximal at internal pH values from 6.5 to 8.0, and decreased by 25% at pH values from 6.5 to 5.5.

The mechanisms involved in the induction of photosynthetic control are complex and not fully understood. It has been known for more than four decades that the accumulation of protons in the lumen can exert a resistance on the H^+^-coupled oxidation of PQH_2_ at Cyt b_6_f [27,35]. However, other observations indicate that under physiological conditions, photosynthetic control could depend on a redox-based mechanism rather than a proton-related mechanism. This view was supported by Johnson and Berry [34], who recently demonstrated that the modulation of the resistance to LEF at Cytb_6_f can occur within milliseconds, incompatible with the slower pH variations. The rapid modulation of Cytb_6_f resistance to LEF was suggested to be the first line of photoprotection, avoiding the overreduction of the PSI acceptor side and the resulting formation of reactive oxygen species. It is possible that both redox- and proton-related mechanisms are complementary, where redox components may alter the pH sensitivity of PQH_2_ oxidation at Cyt b_6_f. This would be similar to the interaction between the carotenoid zeaxanthin and the PsbS protein, which act synergistically to alter the pH sensitivity of energy-quenching qE [46,47]. In that case, PsbS allows plants to respond to light changes within seconds, whereas zeaxanthin requires tens of minutes to fully accumulate, allowing the maximum level of qE for a given light intensity [48].

The results presented here, as those in most of the studies on the effects of pulsed lights on photosynthesis, were obtained with plants grown under constant light (or 60 Hz fluorescent tubes). However, it is important to consider the capacity of plants to acclimate to light fluctuations [3,49,50]. Kanechi [51] showed that net photosynthesis decreased under pulsed light of frequencies equal to or lower than 2.5 Hz (duty ratio of 50% and averaged PPFD of 200 μmol m^−2^ s^−1^) in plants grown under continuous light, but not in plants developed under pulsed lights. Such an acclimation to light fluctuations could result from an enhanced ability of the CEF-PSI following an increase in the concentration of the PGR5 protein that could interact with the PSI complexes [3,52] and/or by faster NPQ kinetics due to higher amounts of the PsbS protein [3,49,50,53].

The ability of plants to perform under pulsed lights may be beneficial for artificial lighting as a recent study has demonstrated that the use of pulsed lights of relatively high frequencies (>100 Hz) and duty ratios (>40%) had no detrimental effects on photosynthetic efficiencies, while allowing modest but significant (10–13%) energy savings [54]. Thus, a better understanding of the functioning of photosynthesis under pulsed light is important to eventually find light conditions that optimize energy consumption and photosynthetic efficiencies.

## 4. Material and Methods

### 4.1. Plant Materials and Growth Conditions

Dwarf sunflower plants (*Helianthus annuus* L., var. Sunspot) were used for the experiments. Plants were grown in a growth chamber under a light intensity of 350 μmol photons m^−2^ s^−1^, with a photoperiod of 16 h/8 h and day/night temperatures of 25 °C/17 °C. Plants were watered daily with tap water and received an appropriate amount of 20-20-20 fertilizer once a week.

### 4.2. Light Treatments and Measurements of CO_2_ Assimilation and Chlorophyll Fluorescence

Before the measurements, plants were transferred from the growth chamber to darkness for about 45 min. Then, working under dim green light, an intact and fully expanded leaf of the third pair from the base of a 5-week-old plant was installed in a modified PLC2 leaf chamber of a LCA4 gas exchange system (ADC Bioscientific, Hoddesdon, UK), supplied with fresh air pumped from the outside with the ADC Leaf Microclimate Control System.

The leaf chamber was placed 20 cm below a 20 × 20 cm^2^ panel of blue LEDs (SL3500 model) connected to a LC-100 light control (Photon Systems Instruments, Drasov, Czech Republic). After 2 min in the leaf chamber under complete darkness, the minimal (Fo) and maximum (Fm) levels of chlorophyll fluorescence (ChlF) were recorded (see below). Then, leaves were exposed for 20 min to a constant light intensity of 1000 μmol m^−2^ s^−1^. After full induction of photosynthesis, leaves were exposed to pulsed lights consisting of 2300 μmol m^−2^ s^−1^ pulses at frequencies ranging from 125 Hz to 0.1 Hz, with a duty ratio of 25% (25% light on, 75% light off), so the averaged light intensity was 575 μmol m^−2^ s^−1^. Leaves were successively exposed to each frequency ranging from 125 Hz to 0.1 Hz and given in a random order, for about 8–10 min during which the rates of net photosynthesis P*n* (integration time of 30 s) were recorded. This time was sufficient for the leaves to reach a new dynamic state, specific for a given frequency. After each frequency, the light was switched off for 2 min to measure the rate of dark respiration. This rate was added to the corresponding rate of net photosynthesis to obtain the rate of apparent photosynthesis.

Simultaneously to the P*n* measurements, ChlF was measured with a XE-PAM fluorometer (Walz, Effeltrich, Germany). ChlF was induced and collected through an optic fiber, fixed at 45° relative to the leaf surface in a holder at the top of the modified PLC2 leaf chamber. When the leaves were still in darkness, the minimal level (Fo) ChlF was measured with the non-actinic measuring light, and then the maximum fluorescence level (Fm) was induced by a saturating pulse (1.5 s duration). After a 20 min exposure to the constant light intensity, ChlF was recorded continuously during the different pulsed lights. Just before the end of the 8–10 min period at a given frequency, a saturating pulse was given to measure the maximum ChlF level, Fm’, emitted by the illuminated leaves. Then, the pulsed light was switched off and far-red light was turned on to measure the minimal Fo’ level.

In addition, simultaneously to CO_2_ assimilation and ChlF, the incident light intensity was measured with a LiCor quantum sensor (model LI-250) connected to a universal transconductance amplifier for LiCor (EME System, Berkeley, CA, USA). Both ChlF and light intensity signals were sent via an acquisition card (DI-158U series, DATAQ Instruments, Akron, OH, USA) to a computer, and the data were recorded at a frequency of 100 Hz.

### 4.3. Calculation of the Photosystem Quantum Yields of the Redox State of the Plastoquinone Pool and of the Conductance of Cytochrome b_6_f to Electron Flow

For each frequency, we obtained the typical ChlF transient during a light pulse by averaging the fluorescence signal of successive light pulses. For the higher frequencies, about 72 cycles were averaged, and only 6 cycles at the lowest frequency (0.1 Hz). To ensure that the successive transients were all synchronized, we also averaged the light signals by taking the onset of the light pulses as a reference point for each light cycle.

For each frequency, we used the mean value of ChlF during the light pulses (excluding the ChlF signals during the dark periods) F*pulse* and the ChlF levels, Fm and Fm’, induced by the saturating pulses to calculate the quantum yield of nonregulated nonphotochemical energy dissipation in PS II, YNO (F*pulse*/Fm), the quantum yield of photochemical energy conversion in PS II, YII ((Fm’ − F*pulse*)/Fm’), and the quantum yield of regulated nonphotochemical energy dissipation in PS II, YNPQ ((F*pulse*/Fm’) − (F*pulse*)/Fm)), according to Klughammer and Schreiber [38].

We also calculated the parameter “1-qL” [1 − [((Fm’ − F*pulse*)/(Fm’ − Fo’)) * (Fo’/F*pulse*)]], which represents the fraction of closed PSII centers [36]. The minimum ChlF levels Fo’ were measured immediately after each period of pulsed light, in the presence of far-red light. As justified by Johnson and Berry [34], this parameter can be used as a good estimate of the reduction state of the plastoquinone pool PQH_2_/(PQ/PQH_2_). Finally, we calculated the apparent conductance of Cyt b_6_f to electron flow by the ratio LEF/(1-qL), where LEF is the product between YII and the light intensity absorbed by PSII (PPFD * 0.5 * 0.85) [34].

### 4.4. P_700_ Redox Kinetics and Measurements of PSI Quantum Yields

A second series of measurements was performed to measure the absorbance changes related to the P_700_ redox state and to estimate PSI quantum yields. Sunflower leaves, grown and pre-treated exactly as described above, were fixed on a reflecting metallic plate, 20 cm below the LED panel. The end of an optic fiber (101-F5) was placed 1–2 cm at 45° from the leaf surface, and connected to a dual-wavelength detector ED P_700_DW (Walz, Effeltrich, Germany). The detector was controlled by a PAM-101 fluorometer. The frequency of the measuring pulse was set at 100 kHz and the damping to the minimum.

Concomitantly to the P_700_ signal, incident light intensity was continuously measured with the LiCor quantum sensor. The absorbance and the light intensity signals were sent via the acquisition card mentioned above to a computer, and the data were recorded at a frequency of 2000 Hz. As for ChlF, the P_700_ signals from successive light cycles were averaged to obtain a typical P_700_ transient during a light period of each frequency.

The PSI quantum yield of photochemical energy conversion (YI) and the quantum yields of nonphotochemical energy dissipation due to donor side limitation (YND) and acceptor side limitation (YNA) were assessed according to the saturation pulse method described by Klughammer and Schreiber [39]. The increase in the P_700_ signal relative to a baseline indicates an accumulation of P_700_^+^. The full P_700_ photooxidation (Pm level) was achieved just after the end of the pulsed light by a 20 ms saturating flash produced by a xenon flash lamp (XF/XMT-103, Walz, Effeltrich, Germany) and given in the presence of far-red light (20 s). On the other hand, the minimum level (Po) was recorded after the complete reduction in P_700_^+^ at the end of the far-red illumination. In the presence of the actinic pulsed lights of different frequencies, the proportion of closed PSI reaction centers due to acceptor side limitation (YNA) was estimated by the difference between the maximum P_700_ signal induced by the Xe flash lamp (Pm’) and the mean value of the typical P_700_ transient during the light pulses Pp relative to the maximum signal (YNA = (Pm’ − Pp)/(Pm − Po)). For YND, it was calculated from the difference between the average P level in the presence of the light pulses Pp and the minimal level (YNA = (Pp − Po)/(Pm − Po)). The values of Pm and Pm’ were estimated by the extrapolation to t = 0 (the onset of the Xe-saturating flash) of the regression calculated from the linear decrease in the P_700_ signal during the first 18 ms of the Xe-flash (see [39], Figure 4)

### 4.5. Kinetics of Analysis

To detect a possible resistance to the electron transfer rate on the donor side of PSI developed at low frequencies of pulsed lights, the kinetics of P_700_^+^ dark reduction at the end of the light pulses were fitted to the equation f = y_0_ + a * exp^−k*t^, where “k” is the reduction rate. The regressions were made with SigmaPlot (v.11.0, Systat Software Inc., San Jose, CA, USA).

### 4.6. Statistical Analysis

All data presented are the means ± S.E.M. calculated from 4 measurements made on different plants. No smoothing was made on the curves used for the calculations of the different PSII and PSI variables. The complete dataset of the results are available in Appendix A.

## 5. Conclusions

The results presented in this study amend the paradigm concerning the decline in photosynthesis under pulsed light of low frequencies. To avoid the overreduction in a pool of photosynthetic electron acceptors during long and intense light pulses, there was a switch from 25 ms to 50 ms after the onset of the light pulses from a productive LEF to a photoprotective CEF-PSI. The cumulative effects of such a CEF-PSI at each repetitive short light pulse resulted in a more acidic lumen, favoring the formation of a large NPQ and of a resistance to LEF at Cyt b_6_f, both decreasing the rate of electron flow toward PSI, and thereby increasing the proportion of harmless P_700_^+^.

## Figures and Tables

**Figure 1 plants-10-02033-f001:**
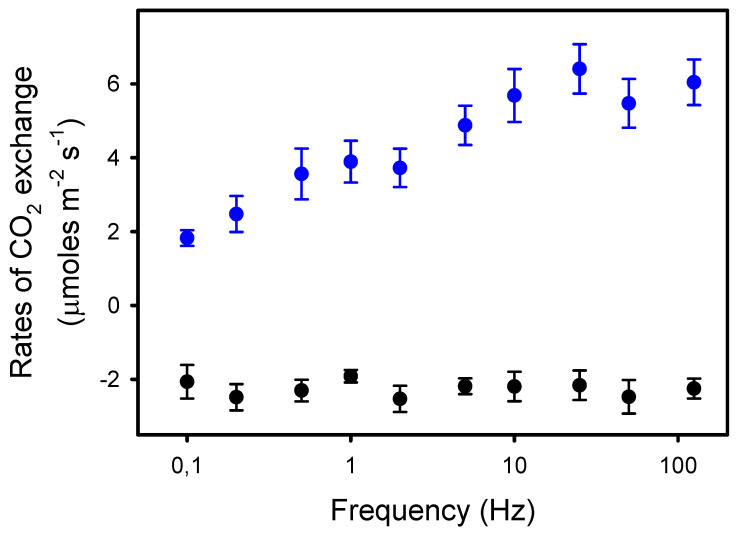
Rates of net photosynthesis P*n* (blue dots) in sunflower leaves exposed to pulsed lights of frequencies ranging from 125 to 0.1 Hz. The duty ratio was 25% and the averaged light intensity was 575 μmoles photons m^−2^ s^−1^. The black dots represent the rates of dark respiration R*d* (-P*n*_dark_) measured between the 8–10 min periods of pulsed lights of different frequencies. The data presented are the means of four independent measurements and the errors bars are the S.E.M. Note the logarithmic scale for the *x*-axis.

**Figure 2 plants-10-02033-f002:**
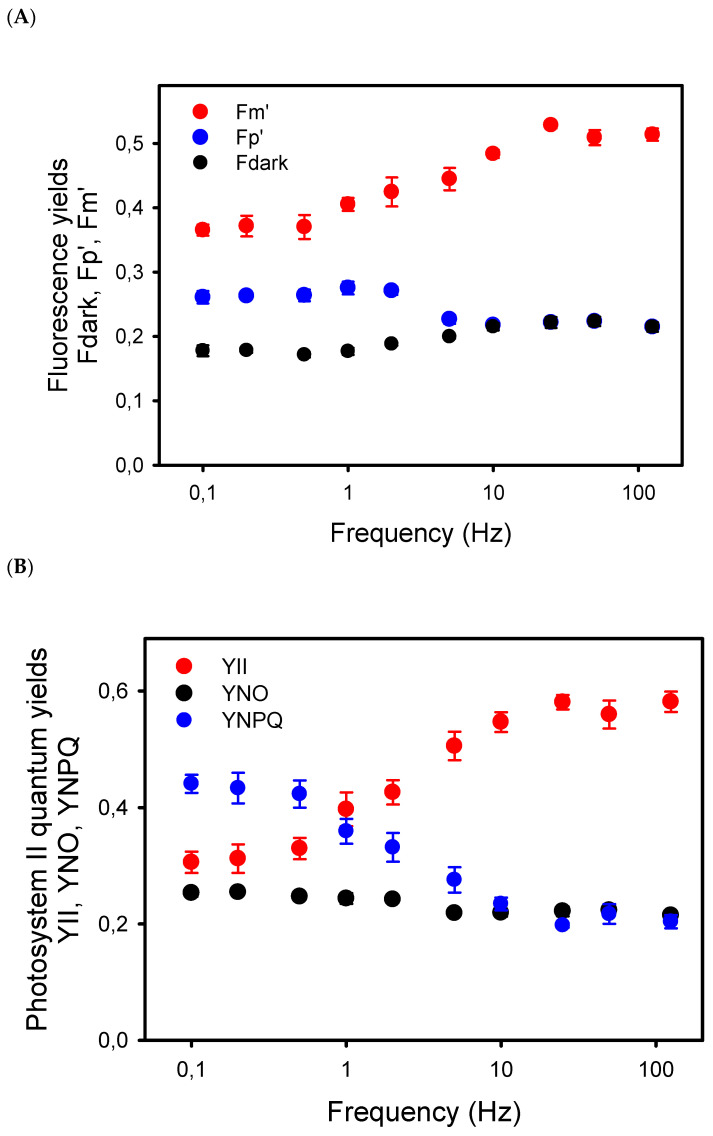
(**A**) Yields of chlorophyll-a fluorescence measured in sunflower leaves exposed to pulsed lights of different frequencies. Fdark represents the mean fluorescence level measured in darkness, i.e., between the light pulses, whereas Fp’ is the peak level of the ChlF reached during the repetitive pulses. Fm’ is the maximum fluorescence level induced by a saturating pulse given just before the end of the 8–10 min period of pulsed light. The Fdark, Fp’, and Fm’ yields are the ChlF values normalized to the maximum ChlF level Fm measured in dark-adapted leaves. (**B**) Effects of the frequencies of pulsed lights on the complementary quantum yields in photosystem II: quantum yield of photochemical energy conversion (YII), and quantum yields of regulated (YNPQ) and nonregulated (YNO) nonphotochemical energy dissipation.

**Figure 3 plants-10-02033-f003:**
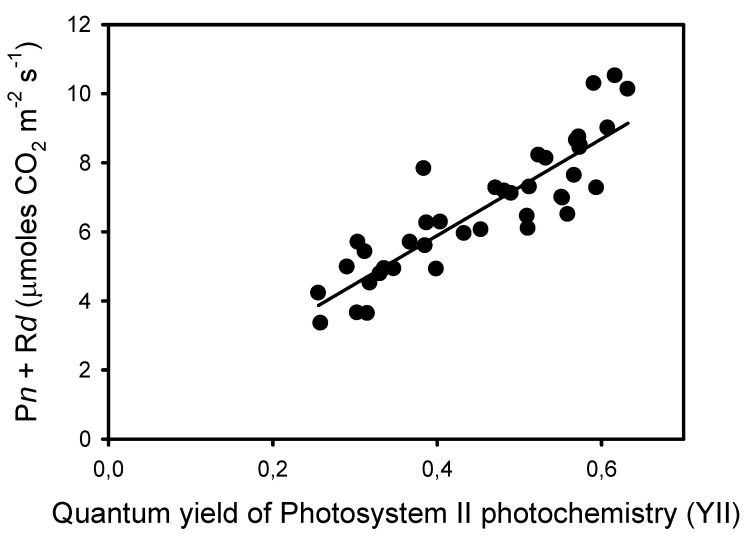
Correlation (R^2^ = 0.780, sum of squares = 28.6, *p* < 0.0001) between the rates of apparent photosynthesis (P*n* + R*d*) and the corresponding quantum yields of photochemical reactions YII measured at ten frequencies and on four leaves from different plants.

**Figure 4 plants-10-02033-f004:**
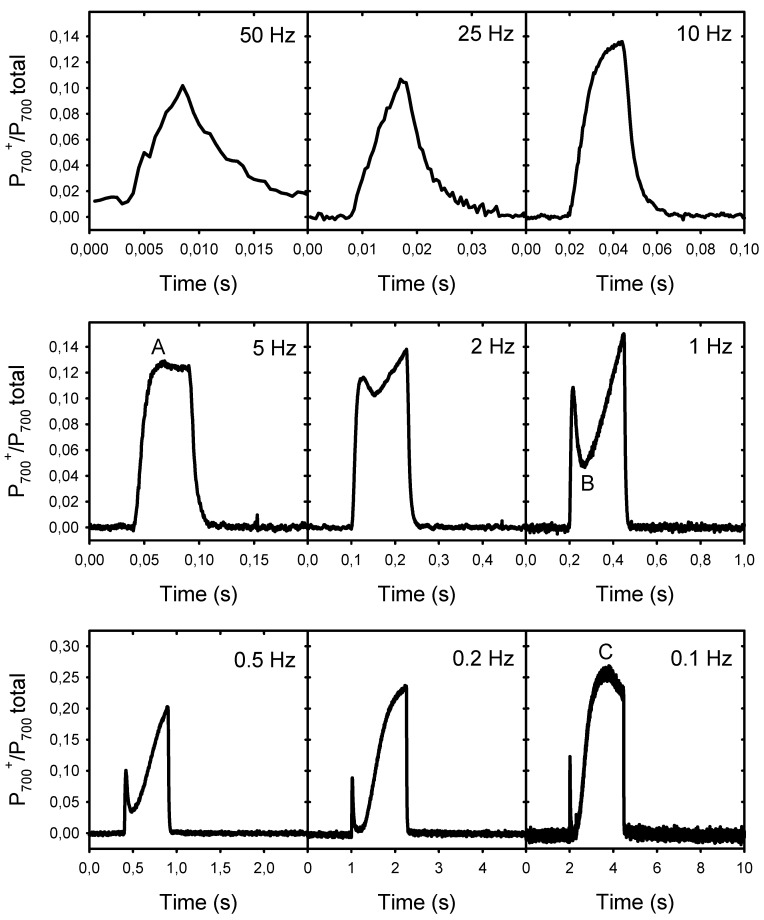
Kinetics of the P_700_ redox changes during the light pulses of different frequencies. The kinetics represent the changes in the proportion of photooxidized P_700_ (from 0 to 1) during one light cycle (25% light, 75% darkness). To facilitate their comparison, the timescales are different. The scales of the *y*-axes are the same for the two upper rows but a different scale was used for the lower row. The letters A, B, and C in the kinetics of 5 Hz, 1 Hz, and 0.1 Hz indicate the distinct phases of the typical kinetics of light-induced P_700_ redox changes.

**Figure 5 plants-10-02033-f005:**
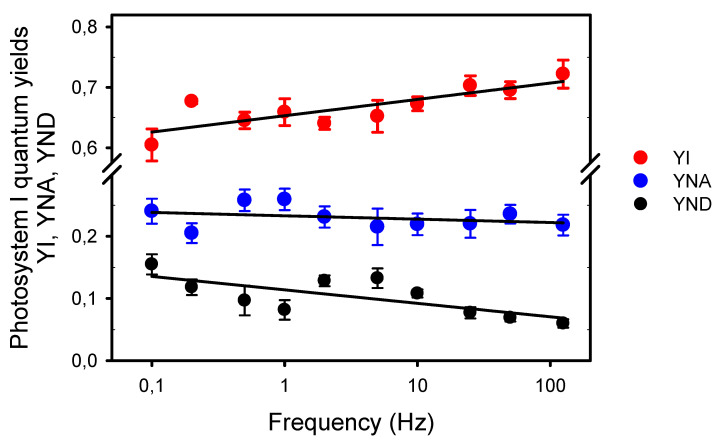
Complementary quantum yields in photosystem I measured under different frequencies of pulsed lights: The quantum yield of photochemical energy conversion, YI; and the quantum yields of nonphotochemical energy dissipation due to donor side limitation, YND, and acceptor side limitation, YNA. The slope a of the logarithmic regression (a * ln(Yx) + b) for YI is 0.012 (R^2^ = 0.670, S.S.E. = 0.003, *p* = 0.0038), −0.002 for YNA (R^2^ = 0.095, S.S.E. = 0.003, *p* = 0.3851), and −0.009 for YND (R^2^ = 0.525, S.S.E. = 0.004, *p* = 0.0178). Note the broken *y*-axis between 0.3 and 0.575.

**Figure 6 plants-10-02033-f006:**
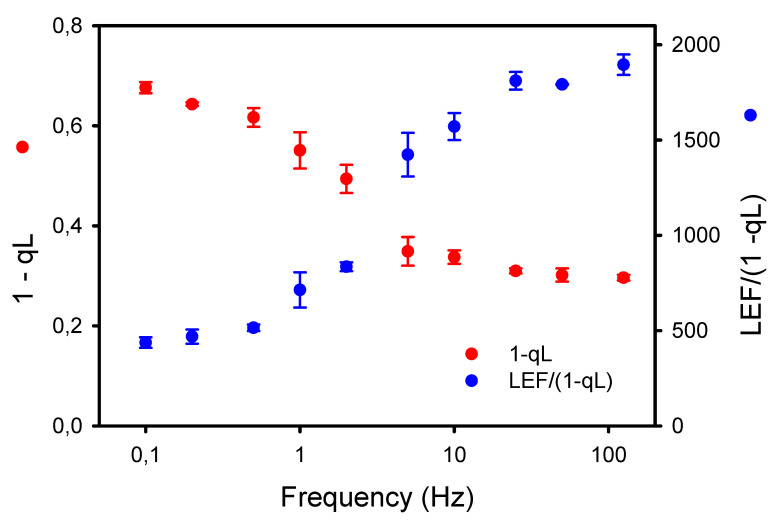
Opposite effects of the frequencies of pulsed lights on the apparent conductance of Cyt b_6_f to linear electron flow (LEF/(1-qL)) and the proportion of reduced plastoquinone PQH_2_ (1-qL) estimated according to the diagnostic method of Johnson and Berry [34].

**Figure 7 plants-10-02033-f007:**
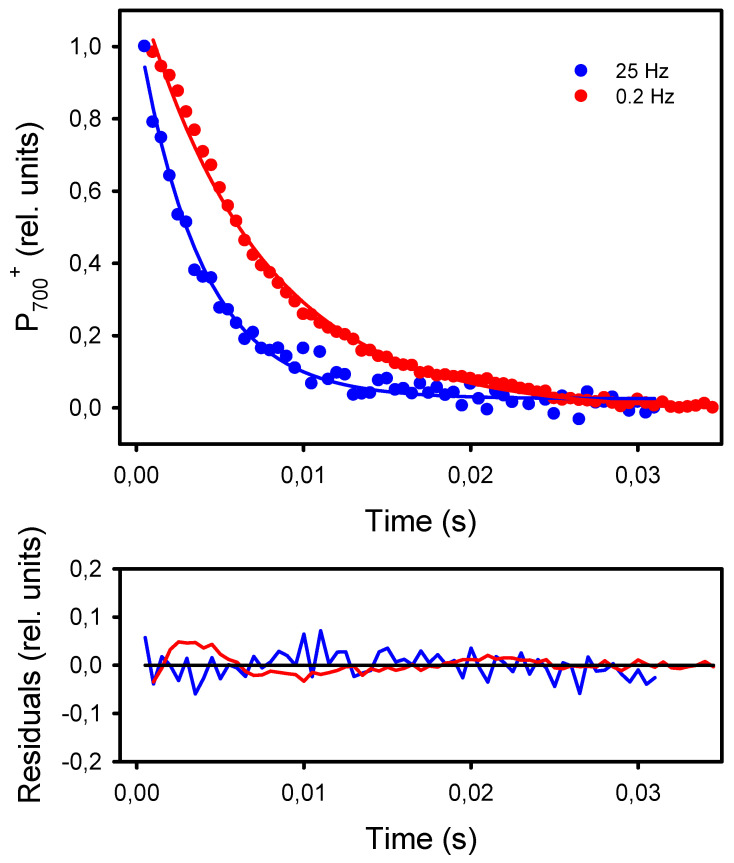
Kinetics of P_700_^+^ dark-reduction observed at the end of the light pulses given at 25 Hz and 0.2 Hz. The kinetics were fitted to the equation f = y_0_ + a * exp^−k*t^, where “k” is the reduction rate. The solid lines represent the curves calculated from the regressions. The residuals of the regressions are shown in the lower graph.

**Figure 8 plants-10-02033-f008:**
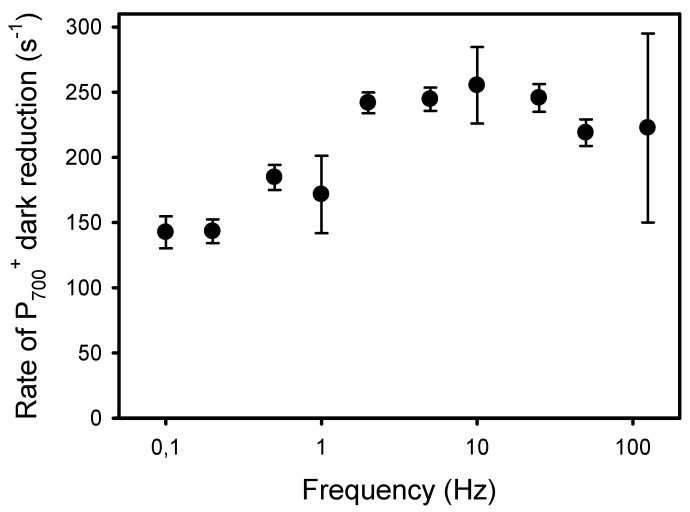
Apparent rate constant (k) of the P_700_^+^ dark reduction occurring at the end of the light pulses given at different frequencies.

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
