# Peer review of "Down-Regulation of Photosynthetic Electron Transport and Decline in CO2 Assimilation under Low Frequencies of Pulsed Lights"

_plants, 2021, doi:10.3390/plants10102033_

Round 1

Reviewer 1 Report

The authors tried to elucidate the molecular mechanisms why net CO2 assimilation decreased at lower frequencies of pulsed lights. Reviewer could not understand the relationship between CEF-I activity and the decrease in net CO2 assimilation rate. Without the information of the redox reactions of P700, PC and Fe/S-signals including ferredoxin, reviewer could not give any explanation about it. Reviewer requires the authors to show the redox state of plastoquinone (1-qL) in Figure 2. NPQ of Chl fluorescence is affected by (1-qL). One short-pulse light can not induce enough delta-pH to support NPQ. What is Fp’?

Author Response

Reviewer 1

1- Reviewer could not understand the relationship between CEF-I activity and the decrease in net CO2 assimilation rate. Without the information of the redox reactions of P700, PC and Fe/S-signals including ferredoxin, reviewer could not give any explanation about it.

We acknowledge that in the first version of the manuscript, we put too much emphasis on CEF-1 without having sufficient observations to support it. One of the major changes made in the revised version is the absence of CEF-I from our conclusion, with the focus on our clear results: the decline of Pn at low frequencies, accompanied by the formation of NPQ and the increase of resistance in the electron transport chain, leading to the larger accumulation of P700+. Near the end of the 3rd paragraph of the revised Discussion, we present four observations (two from our results and two from the literature) from which we can “reasonably suggest that the decline of Pn observed at frequencies lower than 10 Hz is associated to the activation of a CEF-PSI at the expense of LEF supporting carbon metabolism”. We consequently modified the last sentence of the abstract: “Globally, our results indicate that the decline of CO2 assimilation at 10 Hz and lower frequencies coincide with the formation of NPQ and a restriction of electrons flow toward PSI, likely favored by a PSI cyclic electron that contributes to acidifying the thylakoid lumen.” We also thoroughly modified the Discussion (notably the last sentence of the first paragraph), and we removed an entire paragraph (5th par. Od the first Discussion) that was exclusively on CEF-1 (and on CEF-1. S

2- Reviewer requires the authors to show the redox state of plastoquinone (1-qL) in Figure 2. NPQ of Chl fluorescence is affected by (1-qL).

We are grateful to Reviewer 1 for asking us to present the “1-qL” parameter as an estimation of the redox state of the PQ pool. We did measure the Fo’ after each period of pulsed lights (see Material and Methods), allowing us to calculate 1-qL. The results are presented in the new figure 6, and they support the idea of an increased resistance to LEF (presumably at Cyt b6f) resulting in a larger proportion of PQH2. Having 1-qL, it was then interesting to calculate the ratio LEF/(1-qL), the new parameter introduced by Johnson and Berry that estimate the conductance of Cytb6f to LEF.

3- One short-pulse light can not induce enough delta-pH to support NPQ.

We agree that a single short pulse of light cannot decrease luminal pH sufficiently to induce NPQ. But what we observed is the cumulative effect of repetitive light pulses. This is mentioned more clearly in the revised version of the manuscript (throughout the text and more specifically at the 5th line of the conclusion).

4- What is Fp’?

We are very sorry for the confusion created in the first version about Fp’ and Fm’ (there was an unfortunate mistake in the legend of Figure 2). It is now mention three times in the revised version (section 2.2, legend of Figure 2, and 1st paragraphe of the Discussion) that Fp’ is the peak ChlF level reached during the repetitive light pulses. On the other hand, Fm’ is the classical maximum ChlF level induced by a saturating pulse in a light-adapted leaf.

Reviewer 2 Report

In this work, the authors analyze the limitation of CO2 assimilation observed when sunflower plants grown under constant light conditions are subjected to pulsed lights of a frequency lower than 10 Hz. They suggest that this decline is produced by a switch from a productive LEF to a photoprotective  CEF-PSI, which favors the formation of a large NPQ and of resistance to LEF at Cytb6f complex

The manuscript is well written and methods are described clearly. Results obtained are an interesting contribution to the analysis of the different photosynthetic parameters of plants subjected to varying lights.

There is not clear the reason for the CO2 assimilation decline. Do authors suggest that it is because of a decrease of the reduced Fd pool?. A more precise discussion of this point (that appears in the title) should be stated.

Other points:

  • Please, describe more precisely the differences between Fm’ and Fp’. Maybe it should be indicated in the main text rather than in a Figure legend.
  • On page 10, lane 304, authors say that for pulsed lights of low frequencies, LEF is partially restored. Please, explain this affirmation.
  • Same page, lane 324. They say “…to more reduced PSII centers (small YNO increase)…”. Please, explain this sentence. Do authors suggest that a small increase of YNO indicates that there are more reduced PSII centers?

Author Response

Reviewer 2

1- There is not clear the reason for the CO2 assimilation decline. Do authors suggest that it is because of a decrease of the reduced Fd pool? A more precise discussion of this point (that appears in the title) should be stated.

This comment prompted another major modification in the revised version of the MS. We now clearly acknowledge in the 2nd paragraph of the Discussion that we cannot identify undoubtedly the primary cause of the decline of photosynthesis under low frequencies of pulsed light. We now insist this declined is associated to the induction of a photoprotective mode, characterized by the presence of NPQ and an increased resistance in the intersystem electron transport chain, both favoring the accumulation of P700+.  Since we do not have evidences of the primary cause of the photosynthetic decline at low frequencies, we changed our title, from “limitation of CO2 assimilation under low frequencies” to “decline of…”.

2- Please, describe more precisely the differences between Fm’ and Fp’. Maybe it should be indicated in the main text rather than in a Figure legend.

The difference between Fp’ and Fm’ is clearer in the revised version. Please see answer #4 to reviewer 1. 

3-On page 10, lane 304, authors say that for pulsed lights of low frequencies, LEF is partially restored. Please, explain this affirmation.

We apologize for this affirmation that was far from clear. As mentioned in the answer #1 to reviewer 1, we now give much less importance to CEF-PSI in our study. In the previous version, CEF-PSI was part of our conclusion, now we only discuss its likely contribution, by pumping protons into the lumen, to the formation of NPQ and the resistance to LEF, leading to more P700+. So we deleted the paragraph devoted to CEF-PSI and the rapid switch between CEF1 and LEF.

4- Same page, lane 324. They say “…to more reduced PSII centers (small YNO increase)…”. Please, explain this sentence. Do authors suggest that a small increase of YNO indicates that there are more reduced PSII centers?

With the profound changes in the Discussion, this sentence was deleted and now absent in the revised version. In short, larger YNO implies either closed PSII centers (QA-) or damaged PSII (Klughammer and Schreiber 2008). Since the changes of YNO are rapid (during the ChlF transients), this implies that the changes of YNO indicated changes in the proportion of closed PSII centers (QA redox changes).

Reviewer 3 Report

The present manuscript entitled « Down-regulation of photosynthetic electron transport and limitation of CO2 assimilation under low frequencies of pulsed lights “ aims at investigating the response of the photosynthetic electron transfer chain (PETC) and photosynthesis to a brief period of pulsed light using different frequencies. The results show that under low frequencies of pulsed light, CO2 assimilation and the quantum yield of PSII are reduced while the PSII NPQ mechanism is increased (limitation of the electron transfer within the PETC). These observations seemed to be attributed to changes in the P700 oxidized/reduced ratio of the PSI, which arises from a PSI cyclic electron flow. Overall, the purpose of the study can be interesting, but the manuscript needs to be improved in several way to facilitate the reproducibility of the experiments and improve the significance of the results with respect to the PETC and some well-known mechanism (PSI cyclic electron flow, NPQ). Also, the introduction and the discussion parts should be improved and cite more literature (only 36 references for the whole manuscript). In addition, the impact of low frequencies of pulsed light on CO2 assimilation and chlorophyll fluorescence is not depicted as kinetic experiments: how does the CO2 assimilation evolves from the beginning to the end of the experiment ? what for the chlorophyll fluorescence ? These results are important because it could be possible to reach similar results with a longer period of pulsed light at high frequencies. I think that many points need to be clarified and will help to improve the quality of the manuscript and its scientific soundness.

Major points:

  1. Experimental design and kinetics

Many details are lacking regarding the plants/leaves used for the study and the way the experiment was performed. How many leaves are present in a 5-week-old sunflower with your conditions ? Which leaf was used (position from top to bottom) ? Considering the experiment workflow, what was the order for frequencies used for each leaf ? I assume that starting with low frequency could impact on the results with high frequency. In addition, the authors used a 2-min dark step between the different light treatments. How did the authors were sure that this time was enough (Does the NPQ, PSII quantum yield and photosynthesis go back to similar values before any other treatment) ? Another important point is the kinetic of the reduction of CO2 assimilation with respect to chlorophyll fluorescence for low frequencies of pulsed light compared to high frequencies. Indeed, there is often a high NPQ and low PSII quantum yields and net CO2 uptakes during the first minutes of illumination under non-pulsed light because the enzymes of the Calvin-Benson cycle have to be activated, notably by the PETC. So how these parameters evolve specifically in your experiments with high and low frequencies (based on the description in the material and methods, a record every 30 sec was made, so the data are available) ? I think the authors must had some graphs (kinetics for PQII QY, NPQ and An) to the main text to show these results for a very low frequency and a very high frequency (new figure 1) and then to move on the old figure 1 (new figure 2) that elegantly summarizes some important findings (see DOI: 10.1093/jxb/erw054 , DOI: 10.1111/tpj.12945  for non-exhaustive examples).

  1. Measurement of Total CO2 assimilation

The way the Total CO2 assimilation was calculated is highly doubtful for me. First, the measurement of the total CO2 assimilation at light requires to know the quantity of CO2 lost by photorespiratory and respiratory processes (Model of photosynthesis for a C3 plant Farquhar-Von Caemmerer-Berry, which has not been cited at all in the present manuscript while it was used…). In the present manuscript, these parameters were not measured, perhaps because Laisk curves or Co2 response curves coupled to PSII fluorescence measurements are mandatory to approximate these parameters... Second, the dark respiration is higher than the mitochondrial respiration at light (inhibition of glycolysis at light and others processes (see “Berkley Walker, a metabolic flux analysis” https://doi.org/10.1093/plphys/kiab076 for a valuable study)) so there is a high speculation here on the use of dark respiration as a proxy for both photorespiration and light respiration…Third, a recent study showed that net CO2 uptake was highly correlated to Chl and RuBiSCO contents independently of the leaf phenology, and there inherent differences of net CO2 uptake (DOI: 10.3389/fpls.2021.659439 ). In other words, the variation of light respiration and photorespiratory losses on the variation of net CO2 assimilation is very low/negligible even at low photosynthetic intensities. Since the authors measured these two parameters (net CO2 uptake and dark respiration), i think they should show the results for them only. In fact, it would be very informative to see if the dark respiration rate changes with the frequency of the pulsed light treatment (but this change is not expected of course).

  1. Measurement of the Rate of P700+ dark reduction

There are no details about the fitting of the kinetics of P700+ dark reduction with the equation f=y0+a*exp-b*t.  The authors must add some curves with the fitting (as supplemental figures perhaps) and they must show the error sums of squares (SSE) values also, since the R2 is not applicable to non-linear regressions.

  1. Significance of the results

The discussion can be sometimes hard to follow for people outside of this field, and the absence of a scheme summarizing the main findings of the present study with respect to know mechanisms of the PETC is clearly not engaging. I invite the authors to create such figure and cite it appropriately during the discussion.

  1. References

In the present manuscript, many unmissable references from the field were missing regarding photosynthesis, PETC and electron flowsin C3 plants (papers of Graham Farquhar, Suzanne Von Caemmerer, Anja Krieger-Liszkay Pierre Joliot). I think the authors should strengthen the introduction, the material and methods and the discussion with additional literature.

Minor points:

  1. The exact formulas used for the calculation of PSII fluorescence parameters must be stated in addition to the citation of the paper from Klughammer and Schreiber. This will help the reader to reconcile Fm’, F’p and Fdark presented in the figure 2A with the YII, YNO and YNPQ presented in the figure 2B. In addition, there is too much redundancy with these two figures. Please consider to only keep the figure 2B, which is enough informative. Also, since the colors for figures are free-of-charge, can you please add some colors to the figure 2B to facilitate its understanding ?
  2. The scale of the x-axis (Frequency) for many figures seems to be logarithmic. Please specify it in the graphs and in the figure legends.
  3. The correlation in the figure 3 is false from a statistical view. The regression slope and its statistical fitting is calculated with the means of many values. So the exact residual mean sum of squares is falsed/masked. The authors must use all points (please also draw all points and not means with error bars) to give true and reliable results for correlations (see DOI: 3389/fpls.2021.659439 for a non-exhaustive example in the field). Please add the R2 value to the graph. Since this is the ACO2 in this figure, it is expected to show the correlation only for An based on the major point 2.
  4. Please add some colors to the figure 5 to facilitate its understanding. Again here, the R2 values must be calculated with all data points and not means, and the true scores for R2 must be added to the graphs. By the way, based on the legend of the figure 5, why did the authors used a logarithmic regression to fit some points and a classical linear regression to fit other points ? I think the authors must use only linear regressions if they want to compare them and to discuss about their reciprocal changes. In addition, please note that the R2 is not applicable/reliable for non-linear regressions (see DOI: 1186/1471-2210-10-6 ).
  5. The “statistical analyses” part is not enough descriptive. Please specify how the linear regression were calculated (fitting by reducing the sum of squared residues I assume) and which software was used. Usually, it is good to use the software “Rstudio” which is publicly available and is very powerful to perform linear regression modelling (it gives R2 values, F-statistics and p-values associated with the parameters of the model).
  6. Since it is the area of OPEN and FAIR SCIENCE, can you please add some supplementary tables with the raw values (all replicates and not only means+/- SE, since SE is less informative than SD…) used to produce the figures please ? at least a large dataset table is expected to ensure the reusability of the results.

Author Response

Reviewer 3

Major points

  1. Experimental design and kinetics

A- Many details are lacking regarding the plants/leaves used for the study and the way the experiment was performed. How many leaves are present in a 5-week-old sunflower with your conditions ? Which leaf was used (position from top to bottom) ?

Whereas most the leaves in common sunflower plants are arranged alternately along the stem (phyllotaxis 3/8), the first leaves appearing at the base of the plant are arranged in three opposite pairs.  For all the measurement, we used one leaf of the third pair, which was fully developed. There were several younger leaves (4th, 5th…) but decreasing in size and eventually difficult to count, especially in dwarf sunflower plants. This is now mentioned in the revised version (beginning of the 2nd paragraph of section 4.2).

B- Considering the experiment workflow, what was the order for frequencies used for each leaf ? I assume that starting with low frequency could impact on the results with high frequency. In addition, the authors used a 2-min dark step between the different light treatments. How did the authors were sure that this time was enough (Does the NPQ, PSII quantum yield and photosynthesis go back to similar values before any other treatment) ?

As mentioned in the first and revised versions (section 4.2), leaves were first exposed for 20 min to a constant light intensity of 1000 mmol m-2 s-1 to allow the full induction of photosynthesis. Then, leaves were exposed to different pulsed lights, of frequency ranging from 125 Hz to 0.1 Hz, given in a random order, for about 8-10 min during which the rates of net photosynthesis Pn (integration time of 30 s) were recorded.  As shown in an actual trace below, the 8-10 period was sufficient to reach a new “dynamic state”, specific for each frequency, mostly independent from the previous frequency and from the 2-min dark period. Indeed, only the last parts of the fluorescence traces that appeared steady at that time scale were used for the analysis. This minimize the possibility that a ChlF trace at a given frequency influenced by the previous frequencies alters the analysis, the frequencies were given in the different (random) order for the four repetition.

c- Another important point is the kinetic of the reduction of CO2 assimilation with respect to chlorophyll fluorescence for low frequencies of pulsed light compared to high frequencies. Indeed, there is often a high NPQ and low PSII quantum yields and net CO2 uptakes during the first minutes of illumination under non-pulsed light because the enzymes of the Calvin-Benson cycle have to be activated, notably by the PETC. So how these parameters evolve specifically in your experiments with high and low frequencies (based on the description in the material and methods, a record every 30 sec was made, so the data are available) ? I think the authors must had some graphs (kinetics for PQII QY, NPQ and An) to the main text to show these results for a very low frequency and a very high frequency (new figure 1) and then to move on the old figure 1 (new figure 2) that elegantly summarizes some important findings (see DOI: [1-3] , DOI: 10.1111/tpj.12945  for non-exhaustive examples)

We are grateful to Reviewer 3 for sharing with us the idea to characterize the kinetics of photosynthetic induction under constant lights and pulsed lights of different frequencies. Indeed, it would be interesting to compare the decreases of stomatal and biochemical limitations, the activation of photosynthetic electron transport, formation of NPQ… under these different lights. But in a first study, we think that it was important limit the complexity of the system by characterizing some aspects of the “dynamic state” of the photosynthetic apparatus under pulsed lights of different frequencies, and more specifically at frequencies where Pn starts to decrease.

In our experiments, as mentioned above and in the manuscript, the enzymes of the PCR cycle should be sufficiently activated under due to the 20 min pre-illumination (1000 mmol m-2 s-1). Also, their activities should be relatively constant level during the short dark-light cycles. We used only the part of the fluorescence and P700 traces that varied regularly (a kind of steady “dynamic state”). With these different dynamic states, we could say that we a higher degree of complexity than the classical light saturation or A/Ci curves. The next level of complexity would be the induction of photosynthesis from prolonged darkness to pulsed light of different frequencies (including constant) but with the same tatal intensity.

  1. Measurement of Total CO2 assimilation

We are truly embarrassed by this basic but not so uncommon mistake (Wohlfahrt and Lianhong Gu 2016). Effectively, we measured the rates of net photosynthesis (Pn) and of dark respiration (Rd). The sum of these rates corresponds to the apparent photosynthesis. Adding to the latter the rate of photorespiration gives the total or gross photosynthesis. This is now corrected in the revived version. The use of Pn+Rd is important to show a better correlation with YII, extrapolating close to zero (figure 3).

Wohlfahrt and Lianhong Gu (2016) The many meanings of gross photosynthesis and their implication for photosynthesis research from leaf to globe.  Plant, Cell and Environment (2015) 38, 2500–2507  doi: 10.1111/pce.12569

III. Measurement of the Rate of P700+ dark reduction

There are no details about the fitting of the kinetics of P700+ dark reduction with the equation f=y0+a*exp-b*t.  The authors must add some curves with the fitting (as supplemental figures perhaps) and they must show the error sums of squares (SSE) values also, since the R2 is not applicable to non-linear regressions.

Thanks for this constructive suggestion, the kinetics of P700+ dark reduction typical of two frequencies are now shown in the new Figure 7, with the appropriate statistical details.

  1. Significance of the results

The discussion can be sometimes hard to follow for people outside of this field, and the absence of a scheme summarizing the main findings of the present study with respect to know mechanisms of the PETC is clearly not engaging.

I invite the authors to create such figure and cite it appropriately during the discussion.

We rewrote most of the Discussion, hoping to make it clearer and straightforward. Besides the Pn decline at low frequencies, one of the main idea of this study is the induced resistance of electron transport when Pn is limited (here by low frequencies). This resistance at Cyt b6f will receive a lot of attention in a near future considering the impressive model developed by Johnson and Berry, where Cyt b6f plays a key role, as important as NPQ or the activation of Rubisco in the photosynthetic control.

We now close the Discussion by presenting a practical implications of our results, i.e. the possibility to optimize pulsed light for efficient photosynthesis and energy saving.

We strongly considered to present a figure summarizing the functioning of photosynthesis under low frequencies. However, for all reasons mentioned above, it would be presumptuous for to create such a model since we 1) cannot clearly identify the primary cause, the exact origin of the decline of Pn under low frequencies; 2) we have limited data on CEF-PSI (no redox states of Fd, PC), we have no information about the relative implications of the NADPH dehydrogenase and PGR5 proteins in CEF-PSI, and finally, the increased resistance of Cyt b6f to LEF is not induced only by acidic pH in the lumen, but some unclear redox component is also involved. We hope that a clearer picture will soon emerge.

  1. References

In the present manuscript, many unmissable references from the field were missing regarding photosynthesis, PETC and electron flowsin C3 plants (papers of Graham Farquhar, Suzanne Von Caemmerer, Anja Krieger-Liszkay Pierre Joliot). I think the authors should strengthen the introduction, the material and methods and the discussion with additional literature.

By developing the ideas about the resistance of Cyt b6f to LEF, its presence or absence under different physiological conditions, and by better supporting the concept of CEF-PSI, we increased the number of references from 36 to 54.

Minor points:

  1. a) The exact formulas used for the calculation of PSII fluorescence parameters must be stated in addition to the citation of the paper from Klughammer and Schreiber. This will help the reader to reconcile Fm’, F’p and Fdark presented in the figure 2A with the YII, YNO and YNPQ presented in the figure 2B.

Done. Please see section 4.3.

i.b) In addition, there is too much redundancy with these two figures. Please consider to only keep the figure 2B, which is enough informative. Also, since the colors for figures are free-of-charge, can you please add some colors to the figure 2B to facilitate its understanding ?

Figure 2A is important to clearly show that the Fp’ level (now better defined as the peak ChlF intensity reached during the repetitive light pulse, please see answer #4 to reviewer 1) is much lower than the Fm’ level induced by a saturating pulses. This difference is important for the interpretation of the results, as it indicates that the PSII electron acceptors (in fact, the whole photosynthetic electron transport chain) is not fully reduced during the repetitive light pulses.

  1. The scale of the x-axis (Frequency) for many figures seems to be logarithmic. Please specify it in the graphs and in the figure legends.

Done. Please see Figure 1 (but not repeated for the next figures).

iii.        The correlation in the figure 3 is false from a statistical view. The regression slope and its statistical fitting is calculated with the means of many values. So the exact residual mean sum of squares is falsed/masked. The authors must use all points (please also draw all points and not means with error bars) to give true and reliable results for correlations (see DOI: 3389/fpls.2021.659439 for a non-exhaustive example in the field). Please add the R2 value to the graph. Since this is the ACO2 in this figure, it is expected to show the correlation only for An based on the major point 2.

Done. Please see Figure 3.

  1. a) Please add some colors to the figure 5 to facilitate its understanding. Again here, the R2 values must be calculated with all data points and not means, and the true scores for R2 must be added to the graphs.

iv.b) By the way, based on the legend of the figure 5, why did the authors used a logarithmic regression to fit some points and a classical linear regression to fit other points ? I think the authors must use only linear regressions if they want to compare them and to discuss about their reciprocal changes. In addition, please note that the R2 is not applicable/reliable for non-linear regressions (see DOI: 1186/1471-2210-10-6 ).

Done. However, to avoid overcharging the figures with numerous points, we kept the 3 regressions made with the mean values. We provided however more statistical information about the regressions.

The point made by these regressions was to show that the decrease of YI as the frequency decreases is due mostly by an increase of YND, whereas YNA remained more or less constant. The choice to use a classical linear regression or a logarithmic regression is dictated by the data. In Figure 3 for YII vs Pn+Rd, the relation was  clearly linear, and it was important to see it. There is no interest to compare this vs Pn+Rd regression with the changes of YI-YNA-YND vs frequencies.

v).        The “statistical analyses” part is not enough descriptive. Please specify how the linear regression were calculated (fitting by reducing the sum of squared residues I assume) and which software was used. Usually, it is good to use the software “Rstudio” which is publicly available and is very powerful to perform linear regression modelling (it gives R2 values, F-statistics and p-values associated with the parameters of the model).

That will be for a next study. For now, the statistics are pretty simple as we studied only one factor, i.e. frequencies. SigmaPlot is a very efficient software for curve fitting, providing relevant statistical information.

vi.)       Since it is the area of OPEN and FAIR SCIENCE, can you please add some supplementary tables with the raw values (all replicates and not only means+/- SE, since SE is less informative than SD…) used to produce the figures please ? at least a large dataset table is expected to ensure the reusability of the results.

Good idea, we support OPEN and FAIR SCIENCE. A complete database (four repetitions of 25 variables) is now available as supplementary material

Round 2

Reviewer 3 Report

I thank the authors for their answers. Overall, they have adressed the major points and provided a substantial amount of additionnal work that will definitely improve the quality of the manuscript. I still feel a bit sad that no scheme was proposed at the end of the manuscript. However, I really thank the authors for providing the dataset table: it is important to encourage trnasparency and reproducibility in plant science, given the issues encountered by people like Olivier Voinnet.

The manuscript is now suitable for publication !